# A Digital 3D Retrospective Study Evaluating the Efficacy of Root Control during Orthodontic Treatment with Clear Aligners

**Monica Macrì** *, **Silvia Medori, Giuseppe Varvara** and **Felice Festa**

Department of Innovative Technologies in Medicine & Dentistry, University "G. d'Annunzio" of Chieti-Pescara, 66100 Chieti, Italy
* Correspondence: m.macri@unich.it

**Abstract:** This study aimed to investigate the efficacy of torque movement and the incidence of root resorption in the maxillary and mandibular teeth with clear aligner therapy using cone-beam computed tomography. The sagittal root positions, the faciolingual inclinations, and the root lengths of 672 teeth, from central incisors to first molars for each arch, were measured and compared on virtual cross sections from pre-treatment and post-treatment cone-beam computed tomography of 28 patients who received comprehensive orthodontic treatment with clear aligners. An improvement of root position was found in incisors, canines, and premolars of the upper and lower arches: over 78% of their root was centered in the alveolus at the end of orthodontic treatment. There was a statistically significant torque increase for incisors, canines, and first premolars at the end of therapy. The most considerable torque changes were achieved in incisors and canines, while the lowest was in posterior teeth. The maxillary and mandibular central incisors achieved $3.26 \pm 1.95°$ and $2.97 \pm 2.53°$ of mean torque increase, respectively. The root length loss was greater in the upper and lower central incisors. All teeth showed mild resorption (<10%) except for two upper lateral incisors, which showed moderate resorption (10.79% and 10.23%). Comprehensive treatment with clear aligners improved sagittal root position and increased torque, especially in the anterior teeth. Most teeth showed mild resorption after clear aligner therapy, and only two showed moderate resorption.

**Keywords:** 3D; artificial intelligence and health; CBCT; digital health; emerging technologies

## 1. Introduction

Clear aligner therapy, consisting of customized, removable appliances, has been widely used in clinical practice as a more aesthetic and comfortable alternative to multibracket appliances.

In the beginning, the aligners were limited only to mild malocclusions, such as anterior crowding, or to periodontal patients; through the years, thanks to advances in technology and clinical trials, clear aligners have effectively performed major tooth movements, e.g. premolar derotation as well as molar distalization [1]. Despite the predictability of the treatment, its clinical potency remains debatable; opponents have remarked on the need to require mid-course correction or case refinement, especially when treating complex malocclusions, whereas advocates have remained convinced of successful outcomes at the end of the therapy [2].

Compared with our early ancestors, the modern human face reveals a characteristic spatial distribution of bone deposition and resorption. In humans, the anterior portions of the maxilla and mandible's sub-nasal region are more susceptible to surface resorption during development [3,4]. Furthermore, in the sagittal projection of X-ray examinations, clinicians commonly find that the roots of teeth, especially in the anterior region, are positioned against the labial cortical plate. Therefore, it is fundamental to manage the radicular torque and the root position relative to the orofacial cortical plates during orthodontic treatment.

In fixed orthodontic therapy, torque expression depends on several factors: bracket prescription and material (metal or ceramic brackets), inter bracket distance, the vertical

position of the bracket, tooth morphology, and mode of ligation, as well as size and quality of the wire [5]. Studies have reported that slot-arch wire interaction may not produce the 3D control required to express acceptable 3rd-order movement [6,7]. On the contrary, aligners offer customized prescriptions with none of the disadvantages related to bracket design and positioning or tooth morphology, thanks to peculiar features in the cervical region of the clear aligner, such as power ridges.

One of the side effects that can occur during orthodontic treatment is external apical root resorption (EARR). There are few studies on EARR with clear aligners, and their conclusions still need to be made more public [8]. Most studies suggested that the incidence and severity of EARR with clear aligners were lower than those with fixed appliances [9,10], whereas other research [11] found no significant differences between them. Intermittent force, light force, and shorter treatment duration may be reasons for the minimal EARR with clear aligners [8].

Most previous studies used panoramic or periapical radiographs to evaluate EARR, which may cause distortion and overestimate or underestimate the extent of resorption. In recent studies, cone-beam computed tomography (CBCT) has overcome these shortcomings and improved the accuracy of measuring root length [12,13]. CBCT data are highly reproducible [14] and offer excellent sensitivity and specificity [15].

Consequently, this retrospective study aimed to evaluate, by CBCT, the efficacy of torque movement on the upper and lower teeth during clear aligner treatment, and to investigate the incidence and severity of EARR.

## 2. Materials and Methods

The study group included 672 teeth from 28 subjects (17 females and 11 males) treated with clear aligners between April 2017 and January 2022 in the Department of Innovative Technologies in Medicine & Dentistry at "G. d'Annunzio" University of Chieti-Pescara. Ethics approval (number 23) was obtained by the hospital's Independent Ethics Committee of Chieti. The study protocol was drawn following the European Union Good Practice Rules, and the Helsinki Declaration.

The patients' ages are between 18 and 38 years, with an average of 27 years and 6 months. The treatment did not require extractions but did require the dentoalveolar expansion of the maxillary arch to resolve crowding and allow spontaneous mandibular advancement. All subjects requested more than 14 aligners for the arch (comprehensive treatment). Each aligner was changed every 14 days and was worn at least 22 h/day; the average duration of treatment was 25 months.

All subjects met the following inclusion criteria: (1) growth was completed; (2) sufficient height of clinical crown; (3) non-extraction therapy in which the presence of crowding can be managed with protrusion/proclination, expansion, and inter-proximal reduction (IPR); (4) good compliance during treatment; and (5) anti-tightening therapy and physiotherapy with stretching and strengthening exercises of the paravertebral muscles before starting orthodontic therapy.

The exclusion criteria were as follows: (1) systematic disease or drug-taking history affecting tooth movement; (2) orofacial malformation syndromes; (3) periodontal disease; (4) missing teeth (except for third molars); (5) extraction cases; (6) auxiliary treatment during arch expansion stage such as crossbite elastics; (7) surgical case; and (8) previous orthodontic treatment.

At the first visit (T0), records for each participant were collected, consisting of the following: (1) general and dental anamnesis; (2) extraoral and intraoral orthodontic clinical examination; (3) gnathological clinical examination; and (4) visual analogue scale (VAS) and muscular palpation to estimate the pain intensity ratio on patient's face and neck [16].

Each patient underwent a CBCT scan using Planmeca Promax® 3D MID unit (Planmeca Oy, Helsinki, Finland) according to the low dose protocol with these parameters: acquisition time of 15 s, 80 kVp, 5 mA, 35 microSievert (µSv), the field of view (FOV) of 240 × 190 mm, and normal image resolution [17]. The patient's CBCT was performed

with the head oriented according to the Natural Head Position (NHP); the patient was in a sitting position with the back perpendicular to the floor as much as possible. The head was stabilized with ear rods in the external auditory meatus. The patient was instructed to look into their eyes in a mirror 1.5 m in front of them to obtain NHP. The NHP is a physiological and reproducible posture defined for the morphological analysis described in the orthodontic and anthropological literature [18]. Each subject was informed about the radiographic procedure and required to avoid movement and keep centric occlusion with the lip in light contact.

After X-ray scanning, DICOM (Digital Imaging and Communications in Medicine) image files were processed by Dolphin Imaging 3D software (Dolphin Imaging & Management Solutions, Chatsworth, CA) for storage and interpretation. Establishing a predefined patient's head orientation is necessary to obtain a predictable and repeatable three-dimensional (3D) analysis. The skull image was oriented according to NHP in the three planes of space perpendicular to each other, as shown in previous studies [19]: the transverse plane coincides with the Frankfurt plane (FH), a plane passing through two points: Orbital (Or) and Porion (Po); the sagittal plane coincides with the mid-sagittal plane (MSP), a plane perpendicular to the FH plane and passing through two points: Crista Galli (Cg) and Basion (Ba); the coronal plane coincides with the anteroposterior (PO) plane, perpendicular to the FH and MSP, passing through the right and left portion.

After the orientation of the head, the virtual 2D radiograms were extracted in the following sequence:

Lateral teleradiography, on which the cephalometric analysis, according to McLaughlin, is performed

- orthopantomography,
- TMJ stratigraphy,
- cross sections,
- posteroanterior teleradiography,
- superior and inferior submento-vertex
- virtual reconstruction of right and left masseter muscles.

Subsequently, extraoral photos (patient's face in frontal, in the right, and left side views) and intraoral photos (frontal, right, and left lateral photos, and upper and lower occlusal photos) were performed, and the dental arches were scanned using an intraoral scanner, which allows detecting details with an accuracy up to 7 μm.

The virtual setup for each subject was planned, and aligners were manufactured.

At the end of clear aligner therapy (T1), extraoral and intraoral photos, pain assessments (through VAS and muscular palpation), and a CBCT scan were taken for each patient, and 2D virtual radiograms were obtained, as previously described.

For each upper and lower tooth, from right to left, the first molar, the changes in root position, torque, and root length were evaluated by analyzing the cross sections at the start (T0) and the end (T1) of the treatment.

Table 1 shows the number of measurements for the type of tooth of each arch taken into consideration in this study.

**Table 1.** Number of measurements for type of tooth in each arch.

| Tooth | Measurements T0 (n°) | Measurements T1 (n°) | Tot. Measurements (n°) |
|---|---|---|---|
| Central incisor | 56 | 56 | 112 |
| Lateral incisor | 56 | 56 | 112 |
| Canine | 56 | 56 | 112 |
| First premolar | 56 | 56 | 112 |
| Second premolar | 56 | 56 | 112 |
| First molar | 56 | 56 | 112 |
| From central incisor to first molar | 336 | 336 | 672 |

The effectiveness of movement, that is, the evaluation of the sagittal root position in the alveolar bone, was performed by comparing pre-treatment and post-treatment root positions relative to the orofacial cortical plates in the cross sections at T0 and T1 stages.

The sagittal root position was qualitatively evaluated in the midsagittal view according to the rating scale reported by Kan et al. [20] and modified by Aman et al. [21] (Figure 1): In class I, the root is positioned against the labial cortical plate (A); in Class II, the root is centred in the middle of the alveolar housing without engaging either the labial or the palatal cortical plates at the apical third of the root (B); Class III, the root is positioned against the palatal cortical plate (C); Class IV, at least two-thirds of the root is engaging both the labial and palatal cortical plates (D); and Class V, the root is positioned outside the labial cortical plate (E).

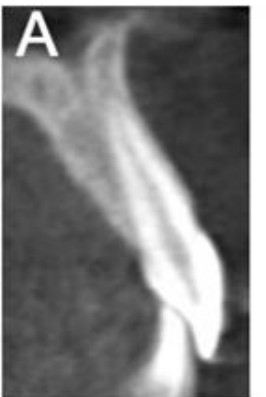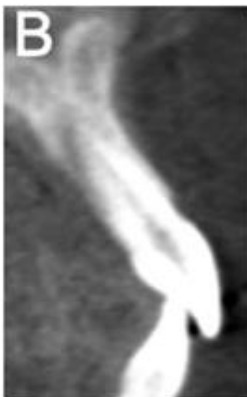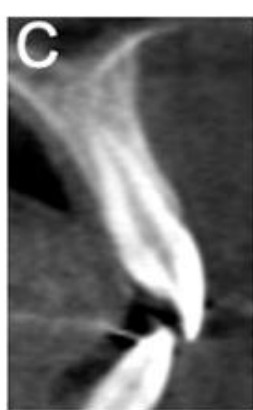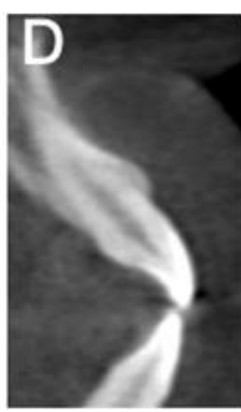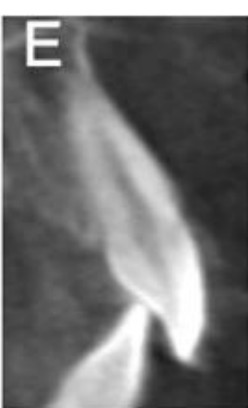

**Figure 1.** Classification of root position relative to cortical plates according to Aman: (**A**), Class I: the root is positioned against the labial cortical plate. (**B**), Class II: the root is centered in the alveolar housing without engaging the labial or palatal cortical plate at the apical third of the root. (**C**), Class III: the root is positioned against the palatal cortical plate. (**D**), Class IV: at least two-thirds of the root engages the labial and palatal cortical plates. (**E**), Class V: the root is positioned outside the labial cortical plate.

Pre-treatment and post-treatment faciolingual inclinations, that is, torque, of each upper and lower tooth from right to left first molar, were measured as the angle formed by the projection of the tooth's long axis on the faciolingual plane and the line of intersection between the faciolingual and mesiodistal planes by the University of Southern California (USC) root vector analysis software program [22] (Figure 2).

If the root center were lingual to the crown center, the torque measurement would be positive; otherwise, it would be negative. The custom USC root vector analysis program uses algorithms to calculate the torque values for all teeth automatically [22]. Subsequently, we reported the torque data in an Excel spreadsheet (version 2019; Microsoft, Redmond, Wash). The difference between pre-treatment and post-treatment torque values for each tooth and the percentage of torque variation were calculated using an Excel spreadsheet.

Additionally, we analyzed the presence of external apical root resorption (EARR). The root length was measured by Dolphin software as the perpendicular distance between the most apical point of the tooth and the reference line at the cementoenamel junction, according to the studies reported by Aman et al. [21] and Jiang et al. [2] (Figure 3).

EARR was defined as a root length loss (mm) between T0 and T1 stages, and the percentage of EARR was calculated as (root length loss/original root length) × 100% using an Excel spreadsheet. The severity of EARR was divided into the following three degrees according to the percentage of EARR: mild (<10%), moderate (10–20%), and severe (>20%).

Two blinded observers, previously instructed to use Dolphin Imaging 3D software, measured the torque and root length values. All measurements were repeated for 10% of the sample after 4 weeks.

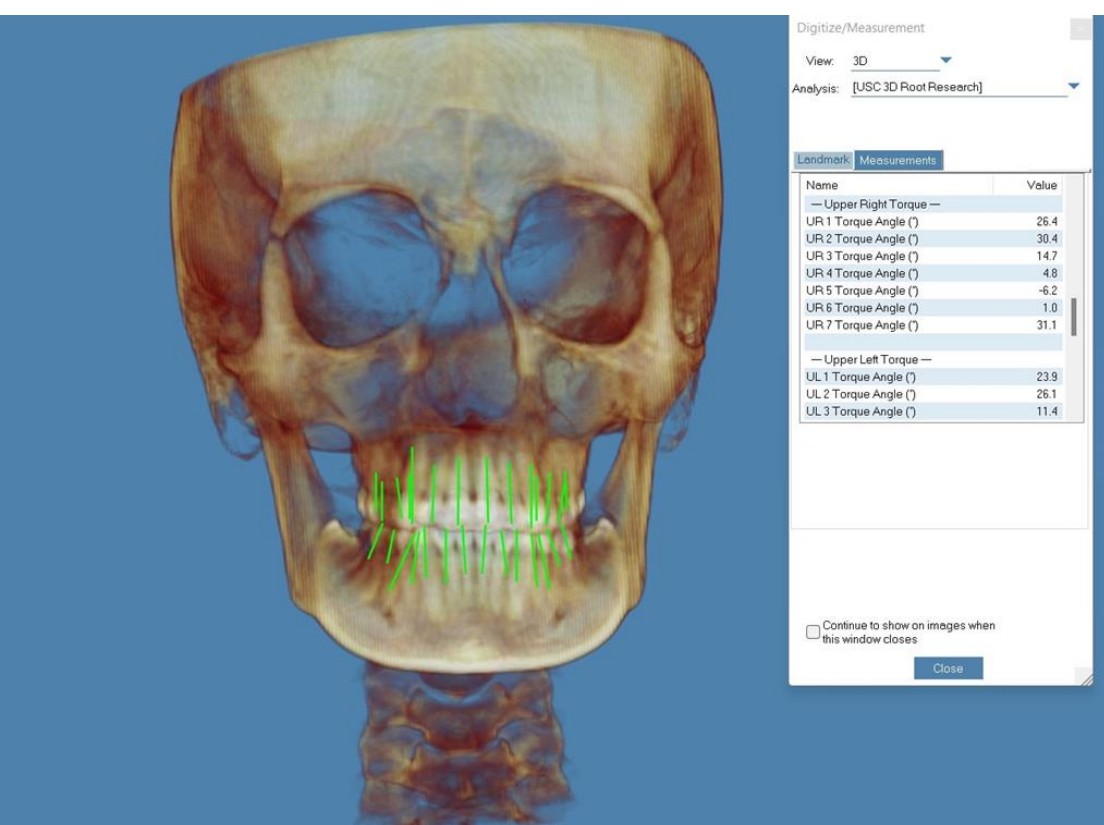

**Figure 2.** Measurements of torque using root vector analysis software program.

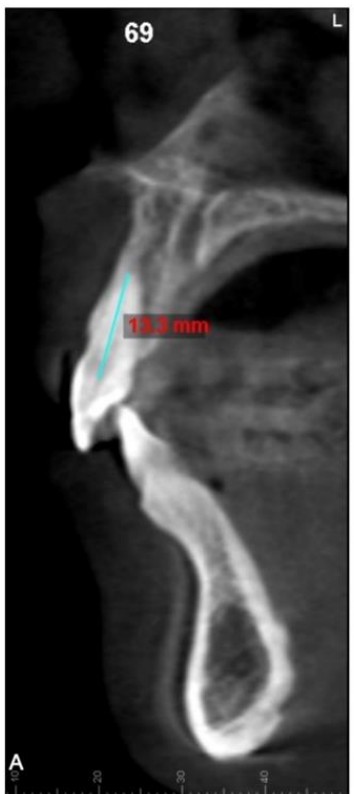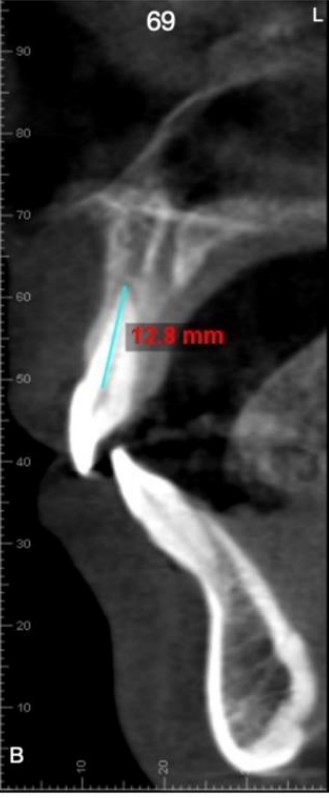

**Figure 3.** Measurements of the root length of the right central incisor on the cross sections: (**A**), pretreatment; (**B**), post-treatment. This tooth shows a slight root resorption of 0.5 mm, equal to 3.7% of the initial root length.

The torque and root length values before and after therapy were subjected to statistical analysis to establish whether torque and root length changes are attributable to orthodontic therapy with clear aligners.

Intraclass correlation coefficients (ICCs) were used to estimate intra-rater and inter-rater reliabilities.

Statistical analyses were performed using Microsoft Excel (version 2019; Microsoft, Redmond, Wash) and StatPlus (AnalystSoft Inc., Walnut, CA, USA). A paired *t*-test was selected to compare torque and root length variations before and after therapy. The level of significance was set at 5%.

### 3. Results

The study analyzed 672 teeth of the upper and lower arches before and after therapy from 28 patients treated with clear aligners.

The ICC tests showed high intra-rater (0.9886 and 0.9883 for torque, 0.9839 and 0.9845 for root length) and inter-rater reliabilities (0.9884 for torque and 0.9880 for root length).

To evaluate the changes in sagittal root positions relative to the orofacial cortical plates, the positions of the roots at the T0 and T1 stages were compared on the cross sections using the classification proposed by Kan et al. [20] and modified by Aman et al. [21].

In the upper and lower arches, no roots were found positioned against the palatal cortical plate or both cortical plates at either T0 or T1 stages.

As regards the upper arch (Figure 4), an improvement in root position has been noticed, especially in the anterior sectors.

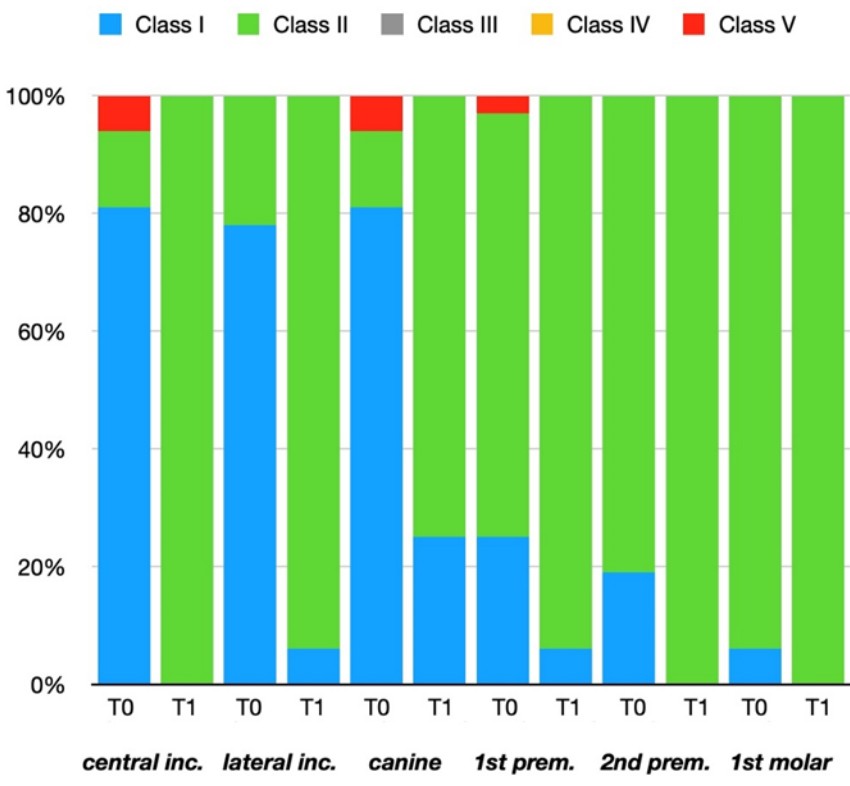

**Figure 4.** Percentage of sagittal root positions before and after therapy in the upper arch.

The central incisors achieved more significant progress. Indeed, all roots were correctly positioned at the end of treatment; the canines showed lower effectiveness since 22% of their roots remained against the labial cortical plate at T1. For the lateral incisors and first premolars, all roots were in Class II al T1, except for only 6% positioned against the buccal cortical plate.

In the second bicuspid, 19% of their roots in class I at T0 were centered in the alveolus at T1. In the first molars, 6% of roots in contact with the labial cortical plate at T0 were found in class II at T1, so 100% of their roots were in the center of the alveolus at the end of therapy.

The aligners satisfactorily managed the roots of the incisors, canines, and first premolars positioned outside the labial cortical plate at T0.

In the mandibular arch (Figure 5), the management of root position was better in the central incisors (from 22% in class II at T0 to 84% in class II at T1). However, 16% of the roots remained against the buccal cortical plate at the end of the therapy.

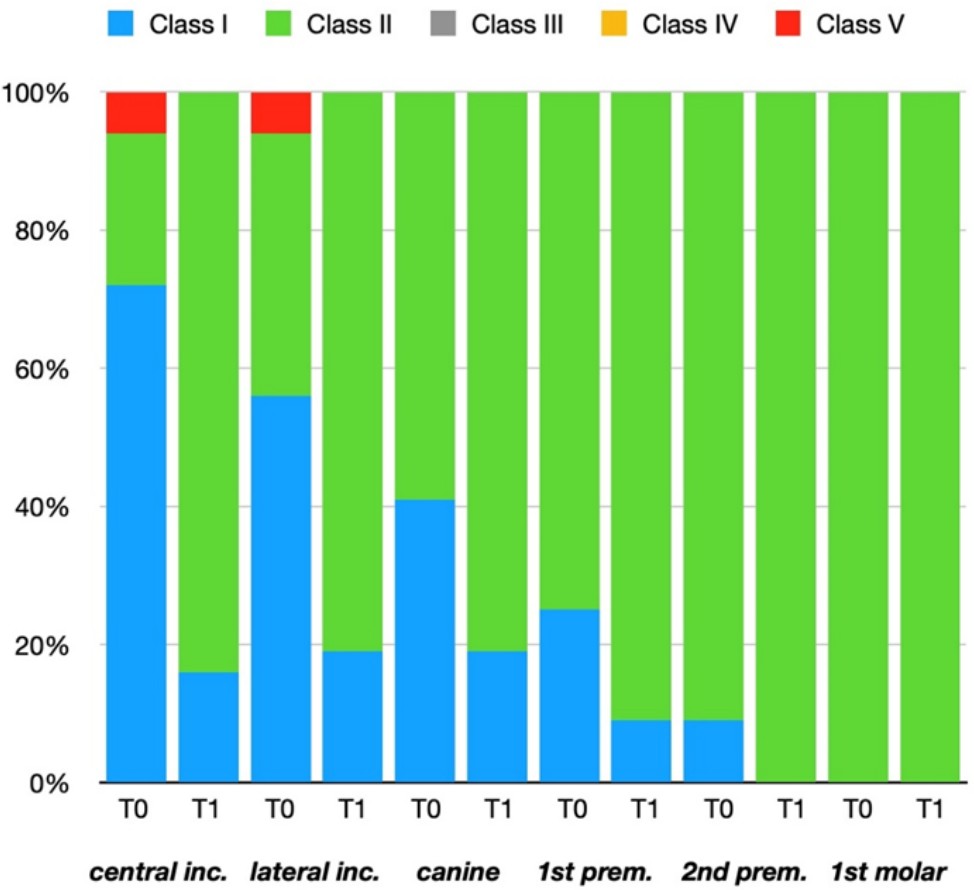

**Figure 5.** Percentage of sagittal root positions before and after therapy in the lower arch.

Regarding the lateral incisors and canines, 81% of the roots were centered in the alveolus at T1, although 19% remained in contact with the labial cortical plate.

The mandibular bicuspids and molars achieved a correct root position at the end of treatment: 91% of the first premolars and 100% of the roots of the second premolars and first molars were positioned between the two cortical plates. We can notice that the initial root position was better in the posterior than in the anterior teeth.

In the lower arch, no roots were found positioned outside the buccal cortical plate at T1.

By comparing Figures 4 and 5, the control of root position after clear aligner therapy was more effective in the upper arch, especially in the anterior sectors.

Subsequently, the torque values before and after therapy were evaluated and compared.

Table 2 illustrates the torque variation of the upper teeth due to the orthodontic treatment. At the end of therapy, the highest mean torque changes were achieved in the incisors, while the lowest mean torque changes were in the posterior teeth.

**Table 2.** Torque variations and paired *t*-test in the upper teeth ($\alpha$ = 0.05).

| Tooth | Torque Variation (°) (Mean $\pm$ SD) | Range (°) | Torque Variation (%) (Mean $\pm$ SD) | Range (%) | Tp | Confidence Interval | *p*-Value |
|---|---|---|---|---|---|---|---|
| Central incisor | 3.26 $\pm$ 1.95 | 9.50–0.10 | 22.82 $\pm$ 12.62 | 66.66–1.20 | 12.49 | 3.78–2.74 | * |
| Lateral incisor | 2.57 $\pm$ 1.86 | 7.50–0.40 | 17.74 $\pm$ 9.09 | 48.68–7.03 | 10.31 | 3.07–2.07 | * |
| Canine | 1.53 $\pm$ 0.89 | 3.90–0 | 14.38 $\pm$ 6.52 | 32.05–0 | 12.84 | 1.77–1.29 | * |
| First premolar | 0.53 $\pm$ 0.36 | 1.10–0 | 8.59 $\pm$ 6.27 | 20.75–0 | 8.24 | 0.65–0.39 | * |
| Second premolar | 0.02 $\pm$ 0.07 | 0.20––0.20 | 0.25 $\pm$ 0.91 | 2.74––2.70 | 1.53 | 0.04–0 | NS |
| First molar | 0.01 $\pm$ 0.05 | 0.10––0.20 | 0.15 $\pm$ 0.73 | 1.49––2.70 | 1.28 | 0.03–0 | NS |

SD indicates standard deviation; and NS is not significant. * $p < 0.05$.

The torque increase was less noticeable in the lateral incisors and canines than in the central incisors. The mean percentage of torque increment varied from 0.15 $\pm$ 0.73% in the first molars to 22.82 $\pm$ 12.62% in the central incisors. This appears to agree with the variations of the root positions at T1 (Figure 4).

In the mandibular arch, the torque variations were the greatest in the central incisors, decreased in the lateral incisors and canines, and were minimal in the premolars and molars, as shown in Table 3.

**Table 3.** Torque variations and paired *t*-test in the lower teeth ($\alpha$ = 0.05).

| Tooth | Torque Variation (°) (Mean $\pm$ SD) | Range (°) | Torque Variation (%) (Mean $\pm$ SD) | Range (%) | Tp | Confidence Interval | *p*-Value |
|---|---|---|---|---|---|---|---|
| Central incisor | 2.97 $\pm$ 2.53 | 9.00–0.20 | 22.63 $\pm$ 11.58 | 67.92–3.17 | 8.78 | 3.65–2.29 | * |
| Lateral incisor | 2.07 $\pm$ 2.16 | 8.10–0 | 12.95 $\pm$ 11.09 | 54.69–0 | 7.15 | 2.65–1.49 | * |
| Canine | 0.94 $\pm$ 0.95 | 3.40–0 | 7.82 $\pm$ 8.16 | 33.33–0 | 7.43 | 1.19–0.69 | * |
| First premolar | 0.18 $\pm$ 0.13 | 0.50–0 | 1.92 $\pm$ 1.67 | 6.25–0 | 7.67 | 0.26–0.13 | * |
| Second premolar | 0.02 $\pm$ 0.07 | 0.10––0.20 | 0.18 $\pm$ 0.79 | 1.41––2.47 | 1.31 | 0.04–0 | NS |
| First molar | 0.02 $\pm$ 0.06 | 0.10––0.10 | 0.21 $\pm$ 0.84 | 1.30––1.43 | 1.65 | 0.04–0 | NS |

SD indicates standard deviation; and NS is not significant. * $p < 0.05$.

The mean increase among mandibular teeth varied between 2.97 $\pm$ 2.53° (central incisors) and 0.02 $\pm$ 0.07°/0.02 $\pm$ 0.06° (second premolars/first molars).

Overall, the torque increase in the lower arch was less than in the upper arch, decreasing from the anterior to posterior sectors.

The results of the paired *t*-test are shown in Tables 2 and 3 for the maxillary and mandibular arches, respectively. In each arch, there was a statistically significant torque increase for the incisors, canines, and first premolars at the end of therapy.

Tables 4 and 5 summarize the differences between the maxillary and mandibular arches' initial and final root lengths.

**Table 4.** Root length changes and paired *t*-test in the upper arch ($\alpha$ = 0.05).

| Tooth | Changes in Root Length (mm) (Mean $\pm$ SD) | Range (mm) | Changes in Root Length (%) (Mean $\pm$ SD) | Range (%) | Tp | Confidence Interval | *p*-Value |
|---|---|---|---|---|---|---|---|
| Central incisor | −0.43 $\pm$ 0.27 | −1.10–0.10 | −3.27 $\pm$ 2.07 | −8.53–0.76 | 8.92 | 0.53–0.33 | * |
| Lateral incisor | −0.53 $\pm$ 0.38 | −1.50–0.10 | −4.03 $\pm$ 2.82 | −10.79–0.78 | 7.88 | 0.66–0.39 | * |
| Canine | −0.14 $\pm$ 0.13 | −0.60–0 | −0.83 $\pm$ 0.75 | −3.57–0 | 6.28 | 0.19–0.09 | * |
| First premolar | −0.08 $\pm$ 0.66 | −0.30–0 | −0.56 $\pm$ 0.42 | −2.20–0 | 8.31 | 0.11–0.10 | * |
| Second premolar | −0.003 $\pm$ 0.02 | −0.10–0 | −0.02 $\pm$ 0.13 | −0.70–0 | 1 | 0.004–0 | NS |
| First molar | −0.003 $\pm$ 0.01 | −0.10–0 | −0.02 $\pm$ 0.12 | −0.68–0 | 1 | 0.004–0 | NS |

SD indicates standard deviation; and NS, is not significant. * $p < 0.05$.

**Table 5.** Root length changes and paired *t*-test in the lower arch ($\alpha = 0.05$).

| Tooth | Changes in Root Length (mm) (Mean $\pm$ SD) | Range (mm) | Changes in Root Length (%) (Mean $\pm$ SD) | Range (%) | Tp | Confidence Interval | *p*-Value |
|---|---|---|---|---|---|---|---|
| Central incisor | $-0.20 \pm 0.28$ | $-0.80$–0 | $-1.58 \pm 1.77$ | $-6.20$–0 | 4.98 | 0.28–0.12 | * |
| Lateral incisor | $-0.09 \pm 0.11$ | $-0.50$–0 | $-0.67 \pm 0.79$ | $-3.60$–0 | 4.71 | 0.13–0.05 | * |
| Canine | $-0.08 \pm 0.07$ | $-0.20$–0 | $-0.50 \pm 0.41$ | $-1.31$–0 | 6.70 | 0.10–0.05 | * |
| First premolar | $-0.003 \pm 0.01$ | $-0.10$–0 | $-0.02 \pm 0.13$ | $-0.71$–0 | 1 | 0.004–0 | NS |
| Second premolar | $-0.003 \pm 0.02$ | $-0.10$–0 | $-0.02 \pm 0.13$ | $-0.75$–0 | 1 | 0.004–0 | NS |
| First molar | $-0.003 \pm 0.02$ | $-0.10$–0 | $-0.03 \pm 0.14$ | $-0.80$–0 | 1 | 0.004–0 | NS |

SD indicates standard deviation; and NS is not significant. * $p < 0.05$.

In the upper arch, the most significant mean root reduction was detected in the lateral incisors ($-0.53 \pm 0.38$ mm), followed by the central incisors ($-0.43 \pm 0.27$ mm) and canines ($-0.14 \pm 0.13$ mm). The root length changes were minimal on the first premolars, while they were negligible for the second premolars and first molars.

According to the severity of EARR, all upper teeth showed mild resorption (<10%), except for two upper lateral incisors, which have undergone moderate resorption (10.79% and 10.23%).

As regards the lower arch, the root length loss was more significant in the central incisors, and then decreased in the lateral incisors and canines and became negligible for the premolars and first molars.

The mean absolute reduction in the mandibular arch varied from $0.003 \pm 0.02$ mm (second premolar and first molar) to $0.20 \pm 0.28$ mm (central incisor).

Neither moderate nor severe resorption was found in any lower tooth, which showed a resorption below 6.20%.

The root length decrease was statistically significant on the maxillary incisors, canines, first premolars (Table 4), and mandibular incisors and canines (Table 5).

## 4. Disscussion

This research aimed to study the efficacy of clear aligners regarding root movement by evaluating the torque before and after treatment and verifying whether the torque variations led to root resorption.

One of the significant challenges for clear aligners is controlling root movement, including the labiolingual inclination, which is more effective in the upper arch than in the lower one. This may suggest that the inter-proximal enamel reduction is a reasonable solution to resolve lower crowding and preserve the roots' integrity.

The torque variation was, as previously reported, higher and statistically significant in the upper and lower incisors: in these teeth, the mean torque variation was more important than $2°$.

The torque increase in the canines was lower than in the incisors: it can be explained by their position at the curvature of the dental arch, which does not allow them to receive optimal force. Moreover, the mechanical efficiency for delivering effective buccally directed force by the aligner decreases from anterior to posterior sectors [23], which is mainly related to more complex root anatomy, thicker cortical plate, higher mastication loading, and greater soft tissue resistance from the cheeks in the posterior region [24].

In literature, the outcomes regarding the torque variation appear to be discordant: this is due to methodological heterogeneity. Some studies analyzed torque without planning for optimized attachments, auxiliaries, or power ridges [25]; others used different materials, even materials no longer used today [1]. Different material properties and aligner production processes affect the force levels and, thus, the predictability of tooth movements [26].

Although clear aligners seem to meet all the criteria of an ideal orthodontic appliance, some biomechanical limits have yet to be overcome [27]. The predictability of tooth movement widely depends on different types of the tooth, dental movement, and arch.

The aligners show the same biomechanical principles as other orthodontic appliances, but the material properties of clear aligners are probably responsible for their weak accuracy in applying torque. Indeed, an aligner's gingival margin is elastic, which would have difficulty controlling forces applied in this region [28]. To overcome this limit, some manufacturers have introduced power ridges, which generate a more significant moment than attachments, as described by Castroflorio et al. [29]: this represents a valid alternative for accurate root torque control, especially on the incisors.

Therefore, the aligners with power ridges may determine a better expression of the torque than can be achieved compared to the multibracket appliances, at least in some prescriptions; moreover, in fixed therapy, Morina et al. [30] showed that various 0.022" self-ligating and conventional brackets on the upper incisors lost an average 10° of torque, known as the so-called "torque play," after the insertion of 0.019" × 0.025" arch wires.

Previous studies [31,32] proved that a clear aligner could produce clinically acceptable results compared to conventional and self-ligating brackets. No statistically significant differences were seen among the three mechanotherapy.

The possibility of managing root movement may help us to correctly plan for the position of the roots within the cortical plates. Only today, the virtual setups have displayed only the coronal changes rather than the radicular ones, which does not accurately reflect the patient's final occlusion.

In addition, there are studies [1] which evaluated the torque only on a small number of aligners and, for this reason, concluded that a greater number of aligners or a torque overcorrection should be required to achieve the desired dental movements. In the present research, the refinements and power ridges have also been included.

It is essential to underline those previous studies concerning torque changes and root resorptions that limited their interest only to the anterior teeth of the upper arch. In the current study, the torque variation and root length loss have been evaluated for both the upper and lower teeth, from incisors to molars.

The subjects selected for this study were adults between 18 and 38 years old. Therefore, the influence of bone metabolism on the movement of teeth during puberty and perimenopause was eliminated.

In our study, comprehensive treatment with clear aligners has led to mild root resorptions; indeed, only two upper lateral incisors showed moderate resorption.

The difference in root length between before and after therapy was found to be statistically significant for all upper and lower anterior teeth; in the posterior sectors, on the other hand, there was no significant root length loss, and this appears to agree with the small variations of root position in the second premolars and molars between before and after therapy.

Data from the literature agree with those found in the current study, which extended its evaluation to the lower arch and the posterior teeth. Liu et al. [33] noticed that most incisors of their sample showed mild or moderate resorption, and only a small percentage (0.625%) underwent severe resorption.

The current study investigated only patients who received a non-extraction treatment; the incidence of EARR was lower in non-extraction than in extraction cases due to the large dental movement and reduction in overjet when closing extraction spaces [34].

Crowding is another risk factor for EARR, with the percentage of root length change in mild crowding significantly lower than in severe crowding.

Root resorption was more related to intrusive and extrusive forces; vertical movement, especially of the anterior teeth, produced greater stress at the root apex [35].

Consequently, the orthodontist should pay attention to the initial torque and tooth movement during the elaboration of the treatment plan.

Regarding the treatment duration, whether this could be a risk factor remains controversial.

In the current study, although the average duration of treatment was 25 months, we found mild root resorption. This can be explained by the intermittent and light force in clear aligner therapy, and by pauses in treatment. At the same time, refinements to a patient's unique prescription are being made, providing the potential for cementum healing [36]. Consequently, the period between therapy initiation and termination does not represent the active treatment duration.

According to the literature, the prevalence of severe resorption with multibracket appliances is higher than in clear aligners [9–37]; fortunately, the long-term survival of teeth with severe root resorption appears to be good.

Regarding the instrumental examination, we used CBCT scan with the low dose protocol because panoramic and periapical radiographs tend to underestimate the finding of root resorption by more than 20% compared to CBCT [38]; indeed, CBCTs show high sensitivity, specificity, and reproducibility.

In orthodontics, the 2D examinations, e.g. panoramic or lateral cephalometric radiographs, lack all information required during diagnosis, treatment planning, and follow-up. In contrast, CBCT scans provide detailed 3D images on skeletal tissue, root resorptions, facial muscle measurement (e.g. masseter), TMJ morphology, and upper airway status.

To limit patient radiation exposure, low-dose CBCT protocols have been proposed. In the current study, we used a low-dose CBCT with a value of the effective dose of 35 μSv [17], considerably lower than that of traditional CBCT.

There is a need for clinical recommendations, guidelines, or position statements from authoritative bodies regarding the use of the low-dose protocol in dentistry. The clinical recommendations issued by the American Academy of Oral and Maxillofacial Radiology (AAOMR) [39] in 2013 concern only high-dose CBCT. In addition, according to AAOMR, in orthodontics, the recommendations are "neither rigid guidelines nor did they represent or imply a standard of care"; therefore, the orthodontist evaluates the use of CBCT based on clinical presentation considering the patient's best interest. Recently, Yeung et al. [40] in their review have underlined that the low-dose CBCT undoubtedly offers greater information both at the beginning and end of the therapy in various dental medicine disciplines, such as in orthodontics to evaluate impacted teeth or alveolar bone quantity and to assess TMJs, in endodontics to detect root fractures, resorptions, and periapical bone loss, and in dental surgery to plan implant insertion and to place temporary anchorage devices.

The current study was subjected to the following limitations. Firstly, we examined only non-extraction treatment. In extraction cases, the management of torque during closing extraction spaces is challenging for the orthodontist [41], and consequently, the root length loss is higher. Secondly, we excluded the use of intermaxillary elastics, so the incidence of EARR was underestimated in our study since the intermittent forces of intermaxillary elastics could lead to greater root resorption. Lastly, we considered only aligners produced by a single manufacturer, and aligner materials' clinical performances vary from manufacturer to manufacturer.

Future papers are suggested to establish whether the materials of other clear aligner manufacturers show similar outcomes to those reported in the present study.

We found, within study limitations, an improvement of sagittal root position and a significant increase in torque in the upper and lower incisors, canines, and first premolars, as well as the highest mean torque changes in anterior teeth. Moreover, these torque increases led to no severe root resorptions, and we observed mainly mild root resorptions following clear aligner treatment.

Therefore, compared to previous papers, our study extended the assessment of root control management to all upper and lower teeth and demonstrated that the clear aligners could enhance the torque in both arches without a substantial root length loss.

## 5. Conclusions

The introduction of clear aligner therapy offers an additional therapeutic tool for the resolution of mild to moderate malocclusions, available for patients, especially adults, who

are attentive to the aesthetics of their smile or who, for personal needs, require little visible and non-altering phonation appliances. Furthermore, the removable aligners allow the maintenance of a correct home and professional oral hygiene.

In the present study, aligners showed an improvement in sagittal root position and torque, especially in the upper anterior region.

To manage the correct root position during clear aligner therapy, the orthodontist should evaluate some factors, such as the staging (movement per aligner), the frequency of aligner changing, the severity of malocclusion, the clinician's experience, the morphology and position of the attachments, and the use of auxiliaries, which could affect the success of the treatment.

Regarding root resorption, most teeth showed mild resorption, and our findings agree with the previous studies. It would be interesting to assess whether the type of attachments used, and the specific tooth movements influence the amount of root resorption.

Finally, in daily practice, it would be desirable to utilize software that, by integrating the three-dimensional data from CBCT, also allows a radicular virtual setup to control root position and prevent apical root resorption and periodontal disease.

**Author Contributions:** Methodology, M.M. and F.F.; Software, S.M.; Formal analysis, M.M.; Investigation, M.M. and G.V.; Data curation, M.M.; Writing—original draft, M.M.; Writing—review & editing, M.M. All authors have read and agreed to the published version of the manuscript.

**Funding:** This research received no external funding.

**Institutional Review Board Statement:** Ethics approval (number 23) was obtained by the hospital's Independent Ethics Committee of Chieti. The study protocol was drawn following the European Union Good Practice Rules, and the Helsinki Declaration.

**Data Availability Statement:** The data presented in this study are available on request from the corresponding author. The data are not publicly available due to privacy.

**Conflicts of Interest:** The authors declare no conflict of interest.

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
