# Peer review of "A Digital 3D Retrospective Study Evaluating the Efficacy of Root Control during Orthodontic Treatment with Clear Aligners"

_applsci, doi:10.3390/app13031540_

Round 1

Reviewer 1 Report

The article is an interesting article using cone-bean CT analysis of the efficiency of torque movement and root resorption incidences during orthodontics. However, the sample size is not sufficient. The article will provide more scientific effects on orthodontics if the sample size increases. More clinical data may improve the quality of this paper. There is also a concern, is this research got IRB proved? Cone-bean CT with higher radiologic risk compared with panoramic film exams. This research should provide IRB proof followed by research ethics. 

Author Response

Dear Reviewer ,

Thank You for your comments and suggestions. We have altered some parts of the manuscript and we hope that it has been improved. We thank You for the input. It was a great pleasure to follow the substantive guidelines

All changes in the revised version of the manuscript are marked with a red and green “text-marker”.

The article is an interesting article using cone-bean CT analysis of the efficiency of torque movement and root resorption incidences during orthodontics. However, the sample size is not sufficient. The article will provide more scientific effects on orthodontics if the sample size increases. More clinical data may improve the quality of this paper. There is also a concern, is this research got IRB proved? Cone-bean CT with higher radiologic risk compared with panoramic film exams. This research should provide IRB proof followed by research ethics. 

Thank you for your revision.

As regards the sample size, the number of study participants was in line with those of other papers concerning the torque evaluation, such as Zhang et. al., 2015, and Zhou et al., 2020 (already cited in the manuscript). The limited sample can be explained by the elimination of confounding factors, such as a previous orthodontic therapy, extraction cases, and use of intermaxillary elastics. In particular, it is likely that patients aged between 18 and 38 years had already undergone a previous orthodontic treatment during childhood. Moreover, children and middle-aged adults have been excluded to eliminate the influence of bone metabolism on the dental movements during puberty and perimenopause. In addition, all patients with low compliance have also been excluded.

Regarding the use of CBCT, we used a CBCT with low-dose protocol to limit patient exposure to radiation. Since the external apical root resorption (EARR) is a three-dimensional topographical change, two-dimensional radiography, such as panoramic examination, have limitations in the accuracy of EARR measurement. Unfortunately, panoramic exams cannot detect root position relative to cortical plates in the three spatial planes. In addition, to achieve a complete orthodontic diagnosis, CBCTs provides information that cannot be examined by 2-D exams.

Best Regards

Reviewer 2 Report

Dear authors, thank you for conducting the present study and for submitting it to Applied Sciences.

Here go a few suggestions:

The study design should be mentioned in the Title

I recommend to place the keyword by alphabetic order.

The Introduction does a good overview. The aim sentence is fine.

May the authors provide the CBCT voxel size?

Were the observers of the CBCT aware of the software used? Why did the authors use the Dolphin Imaging and not the original from Planmeca (Romexis viewer software)? Who performed the imaging? A radiologist?

How many observers conducted the observations? Did the authors conducted intra- and inter-rater tests?

The result look well prepared.

I would like to see in the Discussion a debate regarding the use and applications of use of the CBCT imaging. For instances according to an American association (such as AAE). Are they recommended to accomplish this type of follow ups?

The Discussion lacks the debate of the study limitation, strength, generalization of the results and further studies perspectives.

The authors contribution states an application was done to an ethics committee, however no information regarding that is available in the manuscript.

The references are not in accordance to the journal guidelines.

Author Response

Dear Reviewer 2

Thank You for your comments and suggestions. We have altered some parts of the manuscript and we hope that it has been improved. We thank You for the input. It was a great pleasure to follow the substantive guidelines.

All changes in the revised version of the manuscript are marked with a red and green “text-marker”.

Dear authors, thank you for conducting the present study and for submitting it to Applied Sciences.

Here go a few suggestions:

The study design should be mentioned in the Title

I recommend to place the keyword by alphabetic order.

The Introduction does a good overview. The aim sentence is fine.

May the authors provide the CBCT voxel size?

Were the observers of the CBCT aware of the software used? Why did the authors use the Dolphin Imaging and not the original from Planmeca (Romexis viewer software)? Who performed the imaging? A radiologist?

How many observers conducted the observations? Did the authors conducted intra- and inter-rater tests?

The result look well prepared.

I would like to see in the Discussion a debate regarding the use and applications of use of the CBCT imaging. For instances according to an American association (such as AAE). Are they recommended to accomplish this type of follow ups?

The Discussion lacks the debate of the study limitation, strength, generalization of the results and further studies perspectives.

The authors contribution states an application was done to an ethics committee, however no information regarding that is available in the manuscript.

The references are not in accordance to the journal guidelines.

Dear revisor,

thank you for your revision.

Based on your requests, the changes made to the manuscript are explained below.

The title and the keywords have been modified in the manuscript according to your requests.

Regarding the suggestions concerning the materials and methods, the initial and final CBCT scans were performed by a radiologist in the Department of Innovative Technologies in Medicine & Dentistry at “G. d’Annunzio” University of Chieti-Pescara. A CBCT voxel size of 0.3 mm was selected in our study and was the most chosen in orthodontic field. The voxel size of 0.3 mm was correlated to a normal imagine resolution and to a less ionizing radiation according to the low dose protocol (Caiado GM, Evangelista K, Freire MDCM, Almeida FT, Pacheco-Pereira C, Flores-Mir C, Cevidanes LHS, Ruelas ACO, Vasconcelos KF, Preda F, Willems G, Jacobs R, Valladares-Neto J, Silva MAG. Orthodontists' criteria for prescribing cone-beam computed tomography-a multi-country survey. Clin. Oral Investig. 2022,26,1625-1636 and  Feragalli B, Rampado O, Abate C, Macrì M, Festa F, Stromei F, Caputi S, Guglielmi G. Cone beam computed tomography for dental and maxillofacial imaging: technique improvement and low-dose protocols. Radiol. Med. 2017,122,581-588.).

In our study we used Dolphin Imaging 3D software since this software provides specific tools in orthodontics, compared to Romexis viewer software. In particular, to achieve our aims, we need a reproducible method. For this purpose, Dolphin Imaging 3D software offers the USC root vector analysis program. Moreover, Dolphin Imaging 3D software helps the orthodontist in all treatment phases. For instance, the reliability of Dolphin 3-dimensional voxel-based superimposition and the accuracy of 3-dimensional upper airway evaluation by Dolphin software were well documented in literature (Bazina M, Cevidanes L, Ruellas A, Valiathan M, Quereshy F, Syed A, Wu R, Palomo JM. Precision and reliability of Dolphin 3-dimensional voxel-based superimposition. Am J Orthod Dentofacial Orthop. 2018,153,599-606 and Elshebiny T, Bous R, Withana T, Morcos S, Valiathan M. Accuracy of Three-Dimensional Upper Airway Prediction in Orthognathic Patients Using Dolphin Three-Dimensional Software. J Craniofac Surg. 2020,31,1098-1100.)

The information about observers and intra- and inter-rater tests has been added in “Materials and Methods” and in “Results” sections of the manuscript and indicated in red.

The debate regarding the CBCT use and application has been reported in red colour in the penultimate paragraph in “Discussion” section of the paper.

We apologize for the lack of study limitation, strength, generalization of the results, and further studies perspectives. Thus, in the last paragraph in “Discussion” section of the paper, we reported in red colour your requests.

The references have been re-written according to the journal guidelines.

Best Regards

Round 2

Reviewer 2 Report

Dear author, I have no further concerns.